# Effectiveness of Dupilumab in the Treatment of Patients with Uncontrolled Severe CRSwNP: A “Real-Life” Observational Study in Naïve and Post-Surgical Patients

**DOI:** 10.3390/jpm12091526

**Published:** 2022-09-17

**Authors:** Giancarlo Ottaviano, Tommaso Saccardo, Giuseppe Roccuzzo, Riccardo Bernardi, Alessandra Di Chicco, Alfonso Luca Pendolino, Bruno Scarpa, Edoardo Mairani, Piero Nicolai

**Affiliations:** 1Department of Neuroscience DNS, Otolaryngology Section, University of Padova, Via Giustiniani 2, 35128 Padova, Italy; 2Department of ENT, Royal National ENT and Eastman Dental Hospitals, London WC1E 6DG, UK; 3Ear Institute, University College London, London WC1X 8EE, UK; 4Department of Statistical Sciences and Department of Mathematics Tullio Levi-Civita, University of Padova, 35128 Padova, Italy

**Keywords:** dupilumab, PNIF, nasal cytology, SNOT-22, NPS, ACT, Sniffin’ sticks, PROMs, naïve patients, blood eosinophilia

## Abstract

***Background***: Chronic rhinosinusitis with nasal polyps (CRSwNP) represents 25–30% of all CRS cases, and in the most severe forms it is associated with a poor quality of life and a high rate of nasal polyps’ recurrence after surgery. Dupilumab has been suggested as a treatment option for severe CRSwNP. ***Methods***: Patients with severe CRSwNP receiving dupilumab from January 2021 were followed up at 1, 3, 6, 9 and 12 months from the first administration and were considered for this study. At baseline and at each follow-up, patients underwent nasal endoscopy and completed the Sinonasal Outcome Test (SNOT)-22, a Visual Analogue Scale (VAS) for smell/nasal obstruction, the Nasal Congestion Score and the Asthma Control Test. Peak nasal inspiratory flow (PNIF), a smell test, nasal cytology and blood eosinophilia were also evaluated. ***Results***: Forty-seven patients were included in the study. Of these, 33 patients had a history of previous surgery (ESS) and had recurrent nasal polyps, while 14 patients were naïve to nasal surgery. Both subjective and objective parameters improved after biological treatment and were correlated with each other (*p* < 0.05), except for the SNOT-22 and the nasal polyp’s score. No correlations were found between nasal and blood eosinophilia. No differences were observed when comparing the post-surgical and the naïve groups. ***Conclusions***: Dupilumab improves nasal obstruction and the sense of smell and reduces the level of local inflammation in severe CRSwNP patients in a similar way in both naïve and post-surgical patients.

## 1. Introduction

Chronic rhinosinusitis with nasal polyps (CRSwNP) is a multifactorial disease. It can be associated with genetic disorders, immunodeficiency, anatomical abnormalities and chronic osteomyelitis, but can also be influenced by exposure to environmental factors such as air pollution, smoke, allergens, viruses, bacteria and fungi [1]. CRSwNP represents 25–30% of all chronic rhinosinusitis (CRS) cases and significantly impacts patients’ quality of life [2]. 

In the last years, the treatment of CRS has moved to a new approach to the disease based on the characterization of its immune response and underlying pathophysiological mechanisms, which is known as endotyping. This approach allows for the identification of groups of CRS patients with a higher chance to respond to a specific treatment, thus providing a tailored clinical approach to CRS, which is at the basis of the so-called precision or personalized medicine [3]. In this regard, EPOS 2020 suggested the use of dupilumab, a recombinant human monoclonal antibody that inhibits interleukin-4 and interleukin-13 signaling, in patients affected by type-2 CRSwNP after the failure of surgical treatment [4]. On the other hand, EUFOREA has also extended its indication to naïve patients (those with no previous endoscopic sinus surgery (ESS)) with type-2 CRSwNP [5]. This indication has also been adopted by the Italian Medicines Agency (AIFA) and, today, Italian patients with CRSwNP can obtain access to dupilumab in case of a severe (Nasal Polyp Score ≥ 5 or Sinonasal Outcome Test (SNOT)-22 score ≥ 50) and uncontrolled (patients who did not achieve control of disease with oral corticosteroids (OCS) and/or surgery) disease. Its efficacy and safety have been extensively investigated in two multicenter, randomized, double-blind, placebo-controlled, parallel-group phase 3 trials [6]; however, evidence of its use outside this setting (i.e., in real life) remains poor [6,7,8].

The aim of the present study is to evaluate the efficacy of dupilumab in a cohort of patients with uncontrolled CRSwNP. We focused in particular on the comparison between naïve patients and those who underwent ESS in the past in order to pinpoint any possible differences in treatment response.

## 2. Materials and Methods

### 2.1. Population

The present investigation is an observational study in a real-life setting carried out at the University Hospital of Padua. The study was conducted in accordance with the 1996 Helsinki Declaration and was approved by the Hospital ethical committee (5304/AO/22). Informed consent was obtained from each subject before starting any study-related procedure. 

All patients affected by severe uncontrolled CRSwNP according to EPOS 2020 and receiving dupilumab from January 2021 were included in the study. Dupilumab was administered subcutaneously 300 mg every 2 weeks as an add-on therapy to intranasal corticosteroids (INCS) according to the therapeutic plan set by the AIFA. Inclusion criteria were age of at least 18 years, confirmed diagnosis of diffuse CRSwNP by endoscopy and computed tomography (CT - the last one being not older than 6 months), severe disease stage defined by nasal polyp score (NPS) ≥ 5 or Sinonasal Outcome Tests-22 (SNOT-22) ≥ 50, inadequate symptom control with INCS, failure or intolerance of previous medical treatments (at least 2 cycles of systemic corticosteroid in the last year) and/or failure of previous surgical treatment after ESS with post-operative complications or no clinical benefit. The exclusion criteria were pregnancy, radio-chemotherapy for cancer in the 12 months before the start of the treatment, concomitant long-term oral corticosteroid therapy for chronic autoimmune disorders.

### 2.2. Clinical Evaluation

Patients were evaluated at baseline (before starting the biological treatment) (T0) and at 1 month (T1), 3 months (T3), 6 months (T6), 9 months (T9) and 12 months (T12) from the first administration. At baseline and at each follow-up patients underwent nasal endoscopy using a 0° and/or 30° rigid endoscope and the NPS score was calculated for all of them according to Gevaert et al. [9]. Quality of life was evaluated using the SNOT-22 questionnaire [10], Visual Analogue Scale (VAS) scores for smell and nasal obstruction (NO) [11], the Nasal Congestion Score (NCS) [11] and the Asthma Control Test (ACT score) [12]. Nasal airflow was assessed by means of peak nasal inspiratory flow (PNIF-Clement Clark International) [13], while olfaction by means of Sniffin’ sticks identification sub-test (SSIT) (16 odors) (Burghart Messtechnik GmbH, Holm) [14]. Blood eosinophilia was evaluated at each follow-up, while nasal cytology (as previously described [15]) was performed at T0, T1, T6 and T12 to study nasal inflammatory infiltration. At 12 months after the start of treatment (T12), patients were reassessed and considered eligible to continue the treatment with dupilumab only if all the following criteria were met, as per EUFOREA guidelines: NPS < 4; SNOT < 30; VAS < 5; NCS < 2 [5].

### 2.3. Statistical Analysis

Sample quantiles were used to describe the effect of all relevant variables in time and Bravais–Pearson correlation coefficient to measure the relations between the different indicators. Groups of post-surgical and naïve patients were compared with Wilcoxon test for quantitative variables and with the Fisher exact test for the qualitative ones. For all tests, *p*-values were calculated, and 5% was considered as the critical level of significance. 

The R: a language and environment for statistical computing (R Foundation for Statistical Computing, Vienna, Austria), was used for all analyses [16]. 

## 3. Results

A total of 47 consecutive patients (37 males and 10 women) (mean age: 51.8 years; range 21–74) were included in the study. All subjects reached the 1- and 3-month follow-up (T1, T3), while 35 of them completed the 6-month follow-up and 19 the 12-month follow-up (T12). Fourteen patients had no history of prior ESS (naïve group), while all the rest had history of prior ESS and suffered from nasal polyps’ recurrence (post-surgical group). General characteristics of the whole population, also separated into the post-surgical and the naïve groups, are reported in Table 1. In the post-surgical group (*n* = 33), the mean number of previous ESS was 2.3 ± 1.5, while the mean interval time since last surgery was 73.5 ± 52.3 months. No differences were observed between the post-surgical and the naïve groups at baseline in terms of general characteristics (Table 1). 

Table 2 shows the differences in the Patient-Reported Outcome Measures (PROMs) scores (SNOT-22, VAS scores for smell and NO, NCS and ACT) and the objective measurements (NPS, PNIF, SSIT and cytology findings) between follow-ups (see also Figure 1 and Figure 2).

When we looked at the correlations between the nasal airflow measurements (PNIF) and the reported perception of NO (evaluated by means of VAS-NO), no statistically significant correlation was observed at baseline (T0). However, at T1, T3 and T6, a moderate statistically significant negative correlation was demonstrated (*p* = 0.003, *p* < 0.001 and *p* = 0.043, respectively). No correlations were found at T9 and T12. When we considered the correlations between the measured olfactory function (SSIT) and patients’ reported smell perception (VAS-smell), a statistically significant negative correlation was observed at most of the follow-ups (*p* = 0.04 at T0, *p* = 0.017 at T3, *p* = 0.002 at T6). NPS negatively correlated with SSIT only at T0 (*p* = 0.02). No correlations were found between NPS and SNOT-22, between NPS and PNIF and between nasal eosinophilia (measured at nasal cytology) and blood eosinophilia (Table 3). 

All patients continued their long-term nasal therapy consisting of INCS and nasal douches with saline during the study period [15], and none of them required any OCS courses except for two patients who were considered non-responders to dupilumab (according to EUFOREA criteria [5]) and stopped the biological treatment at the 6-month follow-up. All patients who reached the 12-month follow-up were deemed eligible to continue dupilumab, according to EUFOREA criteria [5].

No adverse events were observed during the treatment period. A transient increase in blood eosinophils was observed in most cases, but only in 3 patients out of 47 did reach values consistent with hypereosinophilia (eosinophils > 1.5 × 10^9^/L). Rapid and spontaneous resolution (within one month) occurred in these three patients without any need for OCS treatment or discontinuation of the biological therapy.

When we compared the post-surgical and the naïve groups, no significant differences in the parameters evaluated were observed at most of the follow-ups with only a few exceptions. The NCS improvement was higher in the naïve group than in the surgery group when comparing the results between T3 and T1 (*p* = 0.01). The smell test (SSIT) showed better scores in the I group than the surgery group when comparing the results between T12 and T9 (*p* < 0.01). Blood eosinophilia showed a significant increase in the post-surgical group when compared to the naïve group when looking at the differences between T9 and T6 (*p* = 0.04) (Table 4).

## 4. Discussion

In the present study, dupilumab was shown to be very effective in the treatment of severe and uncontrolled CRSwNP. A quick improvement in all the objective and subjective (PROMs) parameters was demonstrated in our study group as early as 4 weeks (T1) from the first administration (Table 2, Figure 1 and Figure 2). As a confirmation of the disease control achieved, most of the patients (*n* = 45; 95.7%) did not need any OCS course in addition to their standard treatment (dupilumab + INCS). Moreover, all the 19 patients who completed the 1-year follow-up (T12) showed an adequate CRSwNP control as per EUFOREA criteria (NPS < 4, NCS < 2, VAS < 5, SNOT-22 < 30 and no need for surgery or OCS) [5]. 

When we looked at the subjective perception of nasal obstruction in comparison to the measured nasal airflow (PNIF), no correlation was observed at baseline, corroborating previous data showing a low-to-absent correlation between subjective and objective nasal obstruction [15,17,18,19]. However, a significant correlation became evident after the first month of treatment (T1) and it remained significant at the subsequent follow-ups (T3 and T6). This confirms the reliability of PNIF in the assessment of nasal obstruction [20] and suggests that reported NO becomes more reliable in these patients once the inflammatory component of NO (i.e., sinonasal inflammation) is under control [13,21]. Conversely, a significant correlation between the reported sense of smell, as measured by VAS, and the smell function, evaluated by means of SSIT, was also observed at baseline as well as at T0, T3 and T6. This might be explained by the fact that at baseline (T0) most of the patients were either anosmic or severely hyposmic (SSIT 6 ± 2.7) [22] to the extent that their severe sensory deficit could not fail to be perceived [17]. A lack of statistical significance in the difference between PNIF and reported NO and between VAS-smell and SSIT at T12 could be linked to the low number of patients who reached the 12-month follow-up when compared to the previous follow-ups (T1, T3 and T6). 

Interestingly, when considering the correlation between nasal symptoms (SNOT-22) and NPS score, no correlations were found at baseline or at the other follow-ups. This further confirms the hypothesis that nasal symptoms in these patients are primarily influenced by sinonasal inflammation rather than NP extension [13]. In this regard, when looking into the correlations between NPS score and both the smell test and nasal airflow results, NPS showed various degrees of correlations with SSIT at least at T0 (a marginal correlation was observed at T3 (*p* = 0.099)), while PNIF were not correlated with NPS at baseline or at the other follow-ups.

A reduction in the local nasal inflammation, as measured by means of eosinophil and neutrophil counts, was observed at 1, 6 and 12 months after the start of the treatment with respect to the baseline counts (T0) [23]. No correlation was found when comparing the eosinophilic count at nasal cytology and the blood eosinophilia. This result is probably due to the fact that the majority of the patients included in the study were treated with both INCS and cycles of OCS before starting dupilumab. In fact, corticosteroids have been proven to induce eosinophil apoptosis when given either topically and/or systemically, thus possibly leading to a modification of the eosinophilic count [24]. In addition, no correlations were observed at the other follow-ups, which could be explained by two peculiar features of dupilumab that are interconnected. On the one hand, dupilumab can increase the number of eosinophils in the blood (a transient blood eosinophils increase was observed in most of the cases) [6], while, on the other hand, it reduces the eosinophils’ migration into the tissues [6], hence decreasing the number of eosinophils in the nasal smear (Table 3—see the inverted, yet not significant, correlation shown at T6).

The peculiar composition of our cohort allowed us to perform a comparison between patients with CRSwNP who had previous ESS and those who did not (naïve), with no differences in terms of general characteristics noted at baseline between the two groups. (Table 1) During the pandemic, especially in northern Italy, which was the most hit area from COVID-19 in Italy, nasal surgeries for non-malignant diseases were severely deprioritized. As a consequence, some naïve patients with uncontrolled severe CRSwNP, who satisfied the EUFOREA guidelines [5] to receive dupilumab, were offered biological therapy. Interestingly, no significant differences in subjective and objective measures were observed between the two groups at all follow-ups, thus confirming that both groups obtained a similar improvement while on dupilumab regardless of whether they had received ESS before. It is well known that sinus surgery is not always able to reach long-lasting outcomes in some categories of patients affected by severe CRSwNP, with the polyps’ recurrence rate ranging between 38% and 60% [1,18]. However, because of the high costs of this therapy, indications still remain limited to patients who fail surgical treatment or who are not fit for surgery [23]. 

In the future, the availability of a larger number of alternative molecules in addition to a reduction in the treatment costs might lead to an extension to the current indications to biologic treatment and to the inclusion of naïve patients, especially those with asthma and non-steroidal anti-inflammatory drug (NSAID)-exacerbated respiratory disease, whose CRSwNP is typically more extensive and more recalcitrant to both medical and surgical treatments [25].

The major limitation of the study is the relatively small number of patients included, which prevented any possible comparison between different subtypes of CRSwNP. A reduction in the levels of both eosinophils and neutrophils in the nasal smear was observed in our population. In this regard, it has been shown that neutrophils can represent the dominant inflammatory cell in refractory CRSwNP, regardless of the CRS endotype [26]. In fact, in at least one-third of these patients, not only Asians [27], the inflammation of the nasal mucosa is mainly driven by neutrophils, especially in the most severe forms [28]. The fact that dupilumab is also able to control the level of nasal neutrophils makes it an important treatment option in the management of difficult-to-treat CRSwNP.

In future, larger studies, preferably in a multicenter setting, are warranted in order to allow a stratification of the population by phenotypes/endotypes (i.e., patients with NSAID intolerance and asthma compared to patients with only nasal polyps) and to quantify any difference in treatment response.

## 5. Conclusions

The results of the present study confirm the efficacy of dupilumab in improving nasal obstruction and sense of smell and reducing nasal inflammation severity in patients with uncontrolled CRSwNP. Notably, these effects were equally comparable in both naïve and post-surgical patients. While on treatment with dupilumab, patients reported a remarkable improvement in their nasal obstruction, as reflected by the significant correlation between objective measurements of nasal airflow and the subjective perception of nasal obstruction.

## Figures and Tables

**Figure 1 jpm-12-01526-f001:**
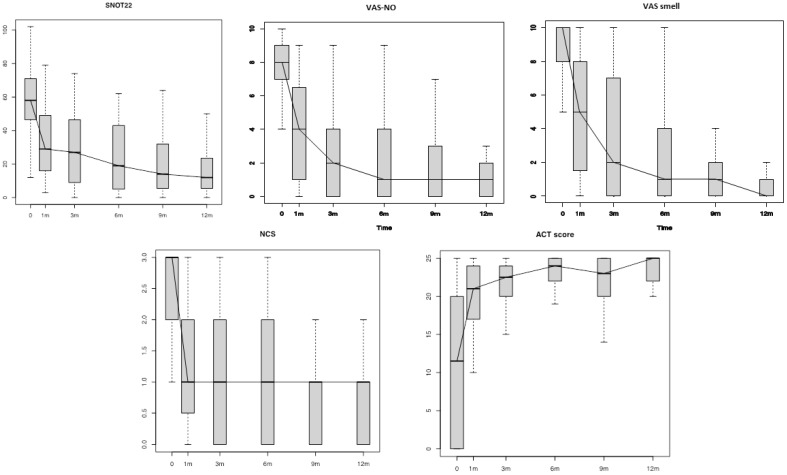
Patient-Reported Outcome measures (PROMs) changes during the study period. SNOT-22: Sinonasal Outcome Test-22; VAS-NO: Visual Analogue Scale for Nasal Obstruction; VAS-smell: Visual Analogue Scale for smell; NCS: Nasal Congestion Score; ACT: Asthma Control Test; m: months.

**Figure 2 jpm-12-01526-f002:**
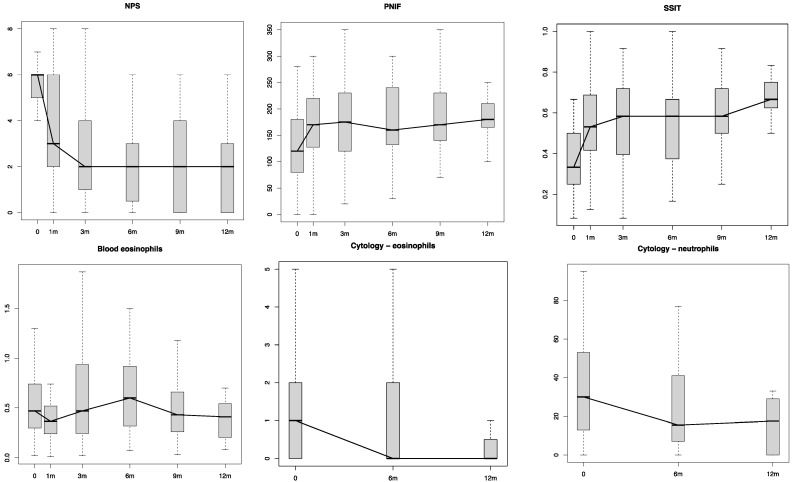
Objective parameter changes during the treatment. NPS: Nasal Polyp Score; PNIF: Peak Nasal Inspiratory Flow; SSIT: Sniffin’ Sticks Identification Test; m: months.

**Table 1 jpm-12-01526-t001:** Patients’ main clinical characteristics, values of the objective measurements and evaluation of symptoms at the baseline for the whole group and for the post-surgical and the naïve groups separately. *P*-value is referred to as the differences between the post-surgical and the naïve groups.

	All *n* = 47	Post-Surgical *n* = 33	Naïve *n* = 14	*p*
**Age, mean (SD), yr**	51.8 (13.5)	53.6 (10.5)	47.6 (18.5)	0.42
**Asthma, n (%)**	25 (53.2)	17 (51.5)	8 (57.1)	1
**Smokers, n (%)**	6 (12.8)	3 (9.1)	3 (21.4)	0.35
**NSAIDs intolerance (%)**	9 (19.1)	7 (21.2)	2 (14.3)	0.70
**Lund–Mackay score**	16.3 (4.2)	16.33 (3.82)	16.23 (5.26)	0.93
**VAS-NO, mean (SD)**	7.62 (2.45)	7.24 (2.66)	8.50 (1.65)	0.13
**VAS-smell, mean (SD)**	8.41 (2.53)	8.47 (2.59)	8.29 (2.46)	0.63
**SNOT-22, mean (SD)**	58.83 (21.53)	59.18 (23.71)	58.00 (15.97)	0.72
**NCS, mean (SD)**	2.47 (0.73)	2.39 (0.67)	2.64 (0.84)	0.09
**ACT, mean (SD)**	9.9 (10.11)	9.92 (9.98)	10 (10.96)	0.97
**NPS, mean (SD)**	5.51 (1.40)	5.33 (1.57)	5.93 (0.73)	0.42
**PNIF, mean (SD)**	128.89 (64.29)	139.52 (63.39)	105.36 (62.03)	0.07
**SSIT, mean (SD)**	6.1 (2.7)	6.0 (2.7)	6.4 (2.8)	0.64
**Blood eosinophilia, mean (SD)**	0.51(0.29)	0.49 (0.30)	0.54 (0.27)	0.61
**Eosinophils cytology, mean (SD)**	2.67 (4.91)	2.04 (3.87)	4.3 (6.93)	0.53
**Neutrophils cytology, mean (SD)**	43.83 (66.37)	54.27 (74.88)	36.7 (20.61)	0.07
**OCS courses/last y, mean (SD)**	2.6 (2.31)	2.19 (1.74)	3.5 (3.13)	0.09

NSAIDs: non-steroidal anti-inflammatory drugs; L-M score: VAS-NO: Visual Analogue Scale for Nasal Obstruction; VAS-smell: Visual Analogue Scale for smell; SNOT-22: Sinonasal Outcome Test-22; NCS: Nasal Congestion Score; ACT: Asthma Control Test; NPS: Nasal Polyp Score; PNIF: Peak Nasal Inspiratory Flow; SSIT: Sniffin’ Sticks Identification Test; OCS: oral corticosteroids; y: year; p: *p*-value.

**Table 2 jpm-12-01526-t002:** Changes in the main clinical outcomes during the study period.

	T1 vs. T0		T3 vs. T1		T6 vs. T3		T9 vs. T6		T12 vs. T9	
	Difference	*p*	Difference	*p*	Difference	*p*	Difference	*p*	Difference	*p*
**VAS-NO**	−4.00	**<0.01**	−1.23	**<0.01**	0.11	0.82	−0.11	0.50	0.17	0.15
**VAS-smell**	−3.66	**<0.01**	−1.52	**<0.01**	−0.50	0.34	−0.48	0.05	−0.22	0.24
**SNOT22**	−27.17	**<0.01**	−4.95	**0.02**	−0.26	**0.01**	−3.64	0.89	0.32	0.92
**NCS**	−1.07	**<0.01**	−0.47	**<0.01**	−0.12	0.30	0.07	0.53	0.00	1.00
**NPS**	−1.98	**<0.01**	−0.74	**<0.01**	−0.34	0.18	0.07	0.86	−0.65	0.08
**PNIF**	39.56	**<0.01**	3.98	0.82	−3.14	0.98	4.48	0.51	−1.84	0.91
**SSIT**	0.18	**<0.01**	0.01	0.90	0.01	0.58	0.02	1.00	0.01	0.75
**Blood eosinophilia**	−0.06	0.16	0.01	0.82	0.10	0.46	−0.10	0.32	−0.09	0.05
	**T1 vs. T0**		**-**		**T6 vs. T1**		**-**		**T12 vs. T6**	
**Eosinophils cytology**	−1.91	0.06	-	-	−0.69	0.34	-	-	2.80	0.37
**Neutrophils cytology**	−21.03	**0.01**	-	-	32.44	0.67	-	-	−40.20	0.10

VAS-NO: Visual Analogue Scale for Nasal Obstruction; VAS-smell: Visual Analogue Scale for smell; SNOT-22: Sinonasal Outcome Test-22; NCS: Nasal Congestion Score; NPS: Nasal Polyp Score; PNIF: Peak Nasal Inspiratory Flow; SSIT: Sniffin’ Sticks Identification Test; *p*: *p*-value. T0: baseline (47 patients); T1: 1 month after the first dupilumab administration (47 patients); T3: 3 months after first dupilumab administration (47 patients); T6: 6 months after first dupilumab administration (35 patients); T9: 9 months after first dupilumab administration (32 patients); T12: 12 months after first dupilumab administration (19 patients).

**Table 3 jpm-12-01526-t003:** Correlations between objective measurements and symptoms during the study period.

	T0		T1		T3		T6		T9		T12	
	corr	*p*	corr	*p*	corr	*p*	corr	*p*	corr	*p*	corr	*p*
**PNIF vs. VAS−NO**	−0.15	0.32	−0.42	**0.003**	−0.51	**<0.001**	−0.34	**0.043**	−0.34	0.07	−0.34	0.15
**SSIT vs. VAS−smell**	−0.31	**0.04**	−0.26	0.08	−0.36	**0.017**	−0.52	**0.002**	−0.10	0.61	0.15	0.54
**NPS vs. SNOT22**	−0.30	0.06	−0.08	0.63	−0.05	0.73	−0.13	0.46	−0.37	0.06	−0.22	0.37
**NPS vs. SSIT**	−0.36	**0.02**	−0.11	0.47	−0.25	0.10	−0.20	0.27	0.05	0.81	0.04	0.86
**NPS vs. PNIF**	0.05	0.76	0.01	0.95	0.08	0.59	0.01	0.96	−0.11	0.57	−0.28	0.26
**Blood eosinophilia vs. eosinophils cytology**	0.17	0.37	0.27	0.21	-	-	−0.49	0.11	-	-	0.87	0.06

PNIF: Peak Nasal Inspiratory Flow; VAS-NO: Visual Analogue Scale for Nasal Obstruction; VAS-smell: Visual Analogue Scale for smell; SSIT: Sniffin’ Sticks Identification Test; NPS: Nasal Polyp Score; SNOT-22: Sinonasal Outcome Test-22; *p*: *p*-value. T0: baseline (47 patients); T1: 1 month after the first dupilumab administration (47 patients); T3: 3 months after first dupilumab administration (47 patients); T6: 6 months after first dupilumab administration (35 patients); T9: 9 months after first dupilumab administration (32 patients); T12: 12 months after first dupilumab administration (19 patients).

**Table 4 jpm-12-01526-t004:** Objective measurements’ and symptoms’ comparison between naïve and post-surgical groups during the study period.

	T1 vs. T0			T3 vs. T1			T6 vs. T3			T9 vs. T6			T12 vs. T9		
	Difference		*p*	Difference		*p*	Difference		*p*	Difference		*p*	Difference		*p*
	Naive	Surg.		Naive	Surg.		Naive	Surg.		Naive	Surg.		Naive	Surg.	
**VAS−NO**	−4.00	−4.00	0.91	−1.38	−1.16	0.66	−0.20	0.24	0.77	0.22	−0.26	0.29	0.12	0.20	0.73
**VAS−smell**	−3.71	−3.64	0.87	−2.38	−1.15	0.36	0.20	−0.79	0.56	−0.44	−0.50	0.98	0.12	−0.50	0.07
**SNOT−22**	−23.36	−28.84	0.46	−7.46	−3.87	0.62	7.00	−3.29	0.82	−8.56	−1.32	0.43	−1.12	1.36	0.11
**NCS**	−0.85	−1.17	0.31	−0.58	−0.43	**0.01**	−0.30	−0.04	0.32	0.22	0.00	0.34	0.25	−0.20	0.06
**NPS**	−1.21	−2.32	0.07	−1.38	−0.47	0.07	1.20	−0.44	0.29	0.11	0.05	0.39	−1.29	−0.20	0.15
**PNIF**	42.86	38.06	0.65	2.31	4.68	0.87	−17.50	2.60	0.37	31.67	−7.75	0.36	−14.38	0.20	0.53
**SSIT**	0.23	0.16	0.26	0.04	0.01	1.00	−0.02	0.02	0.58	0.01	0.02	0.73	0.05	−0.01	**<0.01**
**Blood eosinophilia**	−0.04	−0.07	1.00	−0.03	0.03	0.66	0.29	0.01	0.19	−0.50	0.03	**0.04**	−0.07	−0.11	1.00
**Eosinophils cytology**	−3.60	−1.21	0.37	0.00	−1.10	0.46	0.50	4.33	1.00	−3.60	−1.21	0.37	0.00	−1.10	0.46
**Neutrophils cytology**	−3.10	−28.50	0.07	−7.50	56.40	0.30	−9.50	−60.67	0.40	−3.10	−28.50	0.07	−7.50	56.40	0.30

VAS-NO: Visual Analogue Scale for Nasal Obstruction; VAS-smell: Visual Analogue Scale for smell; Sinonasal Outcome Test-22; NCS: Nasal Congestion Score; NPS: Nasal Polyp Score; PNIF: Peak Nasal Inspiratory Flow; SSIT: Sniffin’ Sticks Identification Test; naïve: group of patients with no previous endoscopic sinus surgery; surg.: group of patients with history of previous endoscopic sinus surgery; *p*: *p*-value. T0: baseline (47 patients); T1: 1 month after the first dupilumab administration (47 patients); T3: 3 months after first dupilumab administration (47 patients); T6: 6 months after first dupilumab administration (35 patients); T9: 9 months after first dupilumab administration (32 patients); T12: 12 months after first dupilumab administration (19 patients).

## Data Availability

The datasets generated and analyzed during the current study are available on reasonable request.

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
