# Peer review of "Effectiveness of Dupilumab in the Treatment of Patients with Uncontrolled Severe CRSwNP: A “Real-Life” Observational Study in Naïve and Post-Surgical Patients"

_jpm, 2022, doi:10.3390/jpm12091526_

Round 1

Reviewer 1 Report

Dear Authors,

Congratulations for an impressive and precious work. I appreciate the structured way you have chosen to convey your goals. 

The study has a real contribution in the CRSwNP treatment, but the limitations of the paper could be mentioned in a much more elaborate manner. 

Author Response

Padova, 7/09/2022

Dear Editor,

we are submitting the revised manuscript “Effectiveness of Dupilumab in the Treatment of Patients with Uncontrolled Severe CRSwNP: A “Real-Life” Observational Study in naïve and post-surgical patients” which was edited according to reviewers’ comments and which we would like you to consider for publication inJ. Pers. Med”.

We would like to thank the reviewer panel for reconsidering our article.

The changes in the manuscript have been highlighted and a list of all changes with a point-by-point reply to the reviewers’ comments has been enclosed.

Best regards,

Giancarlo Ottaviano, MD, PhD

Dept Neurosciences, Otolaryngology Section, University of Padova

Via Giustiniani 2, 35100 Padova, Italy; giancarlo.ottaviano@unipd.it;

fax +39 (0)49 8213113

According to reviewer 1 suggestions:

  1. Limitations of the study have now been discussed in a much more detailed manner (see lines 337-342)

Reviewer 2 Report

Dear Authors,

I found Your work on the effectiveness of Dupilumab in Severe CRSwNP very interesting and clear. I only have a few comments which I detailed below:

- In the Abstract, no mention of the naive and post.surgical group is mentioned util the sentence "No differences were observed when comparing the post-surgical and the naïve groups". I would suggest to revise the abstract in order to make the reader aware of what kind of patients were included in the study;

- Please make sure to explain all acronyms the first time they are used (e.g. NSC on line 94); 

- RESULTS: regarding paragraph at lines 204-208, I would suggest to expand it in order to better present Your results

- In the Results section You state that "When comparing the post-surgical and naïve groups no significant differences in the parameters evaluated were observed at all follow-ups", however in Table 4 I see that there are a few significant differences among the two groups. Please expand this section to better explain the results of Your study;

- In the Discussion section, You state that "A significant reduction of the local nasal inflammation, as measured by means of 278 eosinophil and neutrophil counts, was observed at baseline and after 1, 6 and 12 months 279 after the start of treatment". But how can a significant reduction in local inflammation be detected at the baseline, before the start of the treatment?;

- Please insert a Reference for the sentence at line 287-291.

Overall I found Your work very interesting and well written, Congratulations!

Author Response

Padova, 7/09/2022

Dear Editor,

we are submitting the revised manuscript “Effectiveness of Dupilumab in the Treatment of Patients with Uncontrolled Severe CRSwNP: A “Real-Life” Observational Study in naïve and post-surgical patients” which was edited according to reviewers’ comments and which we would like you to consider for publication inJ. Pers. Med”.

We would like to thank the reviewer panel for reconsidering our article.

The changes in the manuscript have been highlighted and a list of all changes with a point-by-point reply to the reviewers’ comments has been enclosed.

Best regards,

Giancarlo Ottaviano, MD, PhD

Dept Neurosciences, Otolaryngology Section, University of Padova

Via Giustiniani 2, 35100 Padova, Italy; giancarlo.ottaviano@unipd.it;

fax +39 (0)49 8213113

According to reviewer 2 suggestions:

  1. In the Abstract the naive and post-surgical groups have been mentioned before the sentence "No differences were observed when comparing the post-surgical and the naïve groups" (lines 31-32).
  2. All acronyms have now been explained when they first appear in the paper; 

3.      Unfortunately, the paragraph in lines 204-208 is related to figure 2. We believe that due to lines changes the reviewer referred to lines 208-212. Therefore, we expanded this part and rejigged the paragraph (see lines 213-222)

  1. In the Results section, the paragraph regarding the comparison between the naïve and the post-surgical groups has been expanded (see lines 247-254).
  2. We apologize with the reviewer. There was a typing mistake in the sentence which has now been amended (see lines 303-304)
  3. As suggested by the reviewer, reference n. 6 (Bachert et al. Efficacy and Safety of Dupilumab in Patients with Severe Chronic Rhinosinusitis with Nasal Polyps (LIBERTY NP SINUS-24 and LIBERTY NP SINUS-52): Results from Two Multicentre, Randomised, Double-Blind, Placebo-Controlled, Parallel-Group Phase 3 Trials. The Lancet 2019, 394, 1638–1650) has now been added in the sentence (lines 313-314).
